# Surgical Interest of an Accurate Real-World Prediction of Primary Systemic Therapy Response in HER2 Breast Cancers

**DOI:** 10.3390/cancers15102757

**Published:** 2023-05-14

**Authors:** Jose Ignacio Sánchez-Méndez, Mónica Horstmann, Nieves Méndez, Laura Frías, Elisa Moreno, Laura Yébenes, Mᵃ José Roca, Alicia Hernández, Covadonga Martí

**Affiliations:** 1Breast Unit, Obstetrics & Gynecology Department, University Hospital La Paz, 28046 Madrid, Spainahernandezg@salud.madrid.org (A.H.); covadonga.marti@salud.madrid.org (C.M.); 2Department of Obstetrics and Gynecology, Faculty of Medicine, Universidad Autónoma de Madrid, 28029 Madrid, Spain; 3Hospital La Paz Institute for Health Research (IdiPAZ), 28029 Madrid, Spain; laura.yebenes@salud.madrid.org; 4Obstetrics & Gynecology Department, Hospital Clínico Universitario Valladolid, 47003 Valladolid, Spain; 5Breast Unit, Pathology Department, University Hospital La Paz, 28046 Madrid, Spain; 6Breast Unit, Radiology Department, University Hospital La Paz, 28046 Madrid, Spain; mariajose.roca@salud.madrid.org

**Keywords:** breast cancer, HER2 subtype, primary systemic therapy, predictors of response, pathological complete response, magnetic resonance imaging, conservative surgery

## Abstract

**Simple Summary:**

HER2-positive breast cancers are usually diagnosed from clinical findings, and often in a large size, with extensive microcalcifications, and/or with axillary involvement, which may compromise conservative surgery, both in the breast and in the axilla. Nevertheless, HER2-positive breast cancers are usually good responders to neoadjuvant treatment and most of them disappear, although breast microcalcifications do not always do so, nor is an omission of axilla clearance accepted when the initial involvement was extensive. Therefore, it is of great interest to have accessible clinical tools that allow us to determine, with precision, when complete pathological response occurs. In a population of 132 patients with HER2 breast cancers, we have found that the Ki67 value, hormonal receptors condition, and Magnetic Resonance Imaging findings, performed after primary systemic therapy, permit the selection of those patients in whom the benefits of neoadjuvant can safely be transferred to surgical extension.

**Abstract:**

Human epidermal growth factor receptor 2 (HER2)-enriched breast cancers (BC) present the highest rates of pathological response to primary systemic therapy (PST), but they are also the ones that tend to be larger at diagnosis, with microcalcifications and, often, with axillary involvement. If we do not have a reliable method to predict the degree of response, we may not be able to transfer the benefits of PST to surgery. The post-PST surgery planning is guided by the findings in the magnetic resonance imaging (MRI), whose predictive capacity, although high, is far from optimal. Thus, it seems interesting to find other variables to improve it. A retrospective observational study including women with HER2 BC treated with PST and further surgery was conducted. Information regarding clinical, radiological, and histopathological variables was gathered from a total of 132 patients included. Radiological complete response (rCR) was achieved in 65.9% of the sample, and pathological complete response (pCR), according to Miller and Payne criteria, in 58.3% of cases. A higher Ki67 value, the absence of Hormonal Receptors expression, and an rCR was significantly related to a pCR finding. This information impacts directly in surgery planning, as it permits adjustment of the breast resection volume.

## 1. Introduction

Primary Systemic Therapy (PST) is increasingly common in the treatment of breast cancer (BC) because it provides information on the effectiveness of the treatment, has prognostic value [1], and allows a significant number of patients to be offered less aggressive surgical treatments [2,3].

Usually, tumors that show higher rates of response to PST, in terms of reduced tumor burden, are also those that have a more aggressive behavior. These are known as triple-negative BC (estrogen and progesterone receptors, and HER2 negative), and especially HER2 positive BC [4]. The latter account for around 15–20% of all BC cases, and their interest is that, due to development of targeted therapies in recent years, we can offer very effective treatments with controlled toxicity [5]. In this group of tumors, PST combined with anti-HER2 drugs achieves its higher benefits with pathological complete response (pCR) rates over 50%. This fact has made PST the initial therapeutic option in most cases [6,7].

Neoadjuvant therapy in HER2 tumors is based on target drugs (Trastuzumab, Pertuzumab) that act by blocking the HER2 cell signaling pathway, accompanied in the first cycles by some conventional chemotherapy (mainly taxanes [5]). This approach has significantly improved the overall prognosis of these patients. However, approximately 25% of them will present recurrences, which, in most cases, will be in those patients who have not reached pCR after PST.

Nowadays, the best way to predict the pathological response to PST is by imaging techniques, especially magnetic resonance imaging (MRI), although their results are far from optimal [8,9]. Therefore, it is of great interest to identify other predictors of PST response that might be involved in HER2 BC.

The aim of this study is to find which of the factors available in routine clinical practice can accurately predict, in HER2 BC, pCR after PST so that we can translate these results to surgical planning in de-escalation mode, both in the breast and in the axilla, without missing oncologic safety.

## 2. Materials and Methods

A retrospective observational study was carried out by analyzing the population of patients fully treated in our center in the period 2016–2020, with the following:Inclusion criteria: women over 18 years of age, with HER2 immunophenotype BC [10], who received PST (chemotherapy with Trastuzumab with or without Pertuzumab), and subsequently underwent surgery. HER2 status was determined by immune-histochemistry (IHC) using the HER2 polyclonal antibody (DAKO HercepTest) and in situ hybridization by FISH when 2+. Scoring was performed using the current American Society of Clinical Oncology/College of American Pathologists (ASCO/CAP) guidelines [11].Exclusion criteria: patients diagnosed with BC subtypes other than HER2, who underwent primary surgical treatment, who were metastatic, who were males, or who were under 18 years.

The variables studied were age and climacteric status at the time of diagnosis, ethnicity, tumor size per image, focality, laterality, location in the breast, histological differentiation degree (G), hormonal receptors (HR) expression—estrogen receptor (RE) and progesterone receptor (PR)—both in % of positivity, Ki67 value (%) [12], HER2 BC subtype (HR−/HR+), initial axillary clinical involvement (cN0-3), radiological response in the breast (complete/partial/null/progression), radiological response in the breast (%), end-axillary pathological involvement (ycN0-3), breast surgery type (conservative or radical), sentinel lymph node biopsy (SLNB) performed (yes/no), number and degree of positive nodes (ypN0-3), axillary lymph node dissection (ALND) (yes/no), number of positive nodes in ALND, pathological tumor size, degree of pathological response in breast according to Miller and Payne (M&P) [13], Residual Cancer Burden (RCB) [14] criteria, vascular permeation (yes/no), and pathological axillary response degree according to Sataloff criteria [15].

After PST, breast-conservative surgery was performed whenever the residual tumor/breast relationship allowed it. In the axilla, patients without initial axillary involvement, or those cN1 patients who became negative after PST, usually underwent an SLNB, whereas an ALND was performed in those patients cN0 with positive SLNB, in those cN1 patients that did not negativize after PST, or in those with initial cN2-3 axillary involvement.

We performed the descriptive analysis of all variables. Dichotomous or categorical variables were determined by frequencies and percentages. Quantitative variables were analyzed according to whether they followed a normal distribution (Kolmogorov–Smirnov test). Those with a normal distribution were analyzed with the mean and standard deviation, while those not following a normal distribution were analyzed with the median and interquartile range (IQR). We analyzed the possible relationship of the different variables (independents) with the degree of pathological response to PST according to M&P (dependent variable). Comparisons between the two types of variables will be made using contingency tables (chi-square statistic) when compared to a qualitative variable, parametric tests (Student’s *t*) when compared to a quantitative variable with normal distribution, and non-parametric tests (Mann–Whitney U) if it does not follow the normal distribution. Subsequently, we used a binary logistic regression to assess the real influence on pCR of each of the individually significant ones. A *p* value < 0.05 for all variables has been accepted as statistically significant. All this has been done with the support of IBM SPSS Statistic version 26.0 software.

The study was performed in accordance with Good Clinical Practice guidelines and the World Medical Association Declaration of Helsinki. The project was approved by the research ethics committee of our center (PI–5093).

## 3. Results

A total of 132 patients were included. Main patient and disease characteristics are summarized in Table 1 and Table 2.

### 3.1. Clinical Features

Mean age was 52.9 years old (SD: 13.3, 24–81). Half were pre- or perimenopausal. Most of them (60.6%) presented as a localized lesion, with a median total size-measured by MR− of 35.0 mm (IQR 25.0–60.0). Initial clinical axillary involvement was detected in 43.2%. After Caucasians, the second-most-frequent ethnic origin of patients was Asian (Philippines).

### 3.2. Percutaneous Biopsy Histopathological Features

Regarding the histological type, 93.2% were non-special type (NST) invasive BC, histological grade G3 (56.1%), and with a high Ki67 value (58.5%). Regarding the HER2 subtype, the HER2 luminal tumors (HR+) constituted 59.8% (79) of the total, while non-luminal (HR−) constituted 40.2% (53).

### 3.3. Radiological Response after PST

Complete radiological response (rCR) was observed on post-PST MRI in 65.9% (87) of cases.

### 3.4. Surgical Approach

On the breast, conservative surgery was performed in 69.7% (92) of cases. In the axilla, SLNB was performed exclusively in 66.7% (88), SLNB followed by ALND in 15.9% (21), and direct ALND in 17.4% (23).

### 3.5. Histopathological Features Surgical Specimen

In the breast, pCR was confirmed in 58.3% (M&P), a value similar to the RCB criteria one (57.8%) in the 102 cases in which it was applicable. The axilla was negative in 78.8% (104) of the patients: 87.2% of those underwent SLNB, and 39.1% underwent ALND at baseline. Figure 1 shows the management of women according to cN0 or cN1-3, and the results of SLNB and ALND.

#### 3.5.1. Patients without Initial Axillary Involvement-cN0

The SLNB performed on 74 cN0 patients showed metastatic involvement in 8 of them (10.8%), 3 with macrometastases and 5 micrometastases.

#### 3.5.2. Patients with Positive Initial Axillary Involvement—cN1-3

Most cN-positive patients were negative after PST. Metastatic disease was observed in 25 (43.1%) of them, with signs of partial response (N-C) in 75.0%.

### 3.6. Relationship between Variables and the Degree of Response by the M&P System

We considered two groups within the M&P classification system: on one side the G5 response (pCR) and on the other the rest of categories (G1–G4). No significant differences between both groups were found in regard to most of the variables, with the exception of HR expression (ER and PR), Ki67 value, and radiological response. The results are as follows:ER and PR negative status showed higher rates (*p* < 0.001) of pCR (83.0%/41.8%) than the luminal HER2 subtype (Figure 2a).Ki67 Proliferation Index value is directly related (*p* < 0.006) to the pCR rate (Figure 2b).The degree of radiological response had a predictive positive value (PPV) of 78.2% to the pCR rate (*p* < 0.001) (Figure 2c).

### 3.7. Relationship between Pathological Response in the Breast and the Axilla

Overall, 87.5% (28) of patients initially N+, in whom pCR was observed in the breast also presented pCR in the axilla. Similarly, in 82.4% of the patients with ypN0, a complete response was also described in the breast.

### 3.8. Multivariate Analysis

Only three variables remained independent after the multivariate analysis: HER2 subtype (HR+/HR−), Ki67 value, and post-PST radiological response. In this way, the PPV of the MRI reaches 97.3% when considering only HER2+/HR− tumors (IC: 84.2–99.9). In the case of HER2+/HR+ tumors, in which PPV is lower (64.0%), this value reaches 81.5% (CI: 61.25–93.0) when considering tumors with Ki67 levels over 30%.

## 4. Discussion

### 4.1. Clinical Features

In our population, the age variable, unlike what happens in the general population with BC, has a normal distribution and a lower mean, as observed in other studies [3,16]. These findings may be because aggressive cancers occur at younger ages, and because patients with comorbidities that contraindicate PST are concentrated in the group of postmenopausal patients, in whom surgery is usually the primary treatment.

Most of the cases were diagnosed by clinical changes and presented a size over two cm, as published in the Novoa study [3], which may be explained by the fact that the patients are outside the screening age or because they debut as interval tumors.

### 4.2. Percutaneous Biopsy Histopathological Features

Invasive NST was the most common histological type, as it occurs in most BC subgroups [3,17]. Higher histological grade (G3) and higher Ki67 levels were common features among studied specimens, which is consistent with results from other authors [3,18]. Our ratio of HR−/HR+ HER2 tumors is also in line with other publications [3,18].

### 4.3. Radiological Response after PST

Clinical assessment of response to PST is performed using a combination of physical examination and imaging techniques, such as mammography, ultrasound, and MRI [19]. These methods have shown good ability to predict pCR, with MRI being the test of choice for its greater accuracy, especially in HER2-positive and triple-negative tumors, as demonstrated in the meta-analysis of Marinovich et al. [9]. Here, our observed values were also comparable to those of Novoa et al. [3].

### 4.4. Surgical Approach

Most of our patients underwent breast-conserving surgery in a similar rate to the study by Novoa et al. [3], but higher than studies by other investigators [18,20]. The frequent presence of extensive microcalcifications in HER2 tumors often requires a mastectomy, even in the case of rCR [20]. Nevertheless, the need to remove these microcalcifications is controversial as long as they may not imply a worse pathological response and even the in situ component may disappear after the anti-HER2 treatments [20,21].

### 4.5. Histopathological Features of the Surgical Specimen

Due to the different existing criteria to define pCR, comparing results with other studies is sometimes difficult. Some classifications take into account only the absence of the invasive component, whereas others consider the disappearance of the in situ component to be necessary as well. Stricter classifications address not only the response in the breast, but also in the axilla [3,5,22,23]. Our pathological analysis considered the M&P criteria, with the latter addition of RCB. Although the pCR rates are similar between them, the stricter RCB criteria resulted in a slightly lower rate when using this classification. Our data are comparable to other studies using the same chemotherapy treatment regimens (anthracyclines and taxanes) associated with Trastuzumab and Pertuzumab [3], but lower than Vega et al. [18], which reached up to 70% with schemes that include carboplatin.

The drawback of M&P Is that it does not consider the size of the tumor and does not evaluate axillary lymph node involvement (although in practice, the classification of the Sataloff system is used for this). The RCB scale simultaneously evaluates breast and axilla response and is the only one validated as an independent prognostic factor for survival [1,17]. Thus, numerous studies have shown that achieving pCR is the most important parameter for improving overall survival [3,17,24], especially in patients with triple-negative and HER2-positive tumors [3,5].

In the axilla, the percentage of patients who were clinically negative (cN0) at diagnosis, but showed some degree of involvement in the SLNB, was lower than that collected by the literature, and are reflective of, rather than progression during PST, the limitations inherent in radiological evaluation.

More than half of patients with initial axillary involvement did not have axillary disease after PST. This is one of the great advantages of the neoadjuvant therapeutic approach in these cases, because it allows to avoid unnecessary ALND.

### 4.6. Relationship between Pathological Response in Breast and Axilla

We found no significant differences depending on whether the tumors were luminal or not. Other authors [25], on the other hand, found that the association is especially true in HR cases.

### 4.7. Relationship between Variables and Degree of M&P Response

Properly identifying, from the clinical variables available at the time of surgical planning, the subgroup of patients with high pCR rates would allow us to correctly indicate less extensive surgeries both in breast and axilla [5], even when breast microcalcifications persist. This is of great importance in order not to overtreat surgically after PST.

In our study, we have seen relationships in the three variables mentioned—HER2 subtype, Ki67 value, and degree of radiological response measured by MRI—but in the literature, there is no agreement, except in the last one, about which clinical-pathological parameters can be used as predictors of response to treatment [17].

HER2 subtype. Our pCR rate was higher than that of Novoa et al. [3], but only in non-luminal HER2 tumors (RH negative). This has been seen in other series [18,26] and is collected in the meta-analyses of Cortazar or Broglio [24,27], which may be due to a cross-resistance between the HER2 and estrogen receptors pathways, which will confer some resistance to therapies aimed at each of them [28].

Likewise, several studies indicate that the level of amplification of the HER2 gene is a factor related to achieving a pCR, so that the greater the intensity in amplification, the greater the number of pCR, and the greater benefit in DFS and OS [3,29]. Finally, in the GeparQuattro study, it was observed that the rate of pCR was higher in tumors with overexpression of p95, without being clear whether this relationship is due to better response to chemotherapy or trastuzumab [30].

KI67. In our study, the value of Ki67 is directly related to the degree of pathological response in the breast (*p* = 0.001), in the sense that most patients with high Ki67 achieved pCR. There are numerous studies [5,18,31] with similar conclusions, although the problem of this marker is related to its low reproducibility.Radiological response after PST. In the daily clinic, MRI is accepted as the gold standard for predicting the degree of response to PST [9], although its PPV does not exceed 80%. Therefore, it would be very useful to have other variables that improve it. In our study, we found that variables such as Ki67 (high) and HER2 subtype (non-luminal HER2) improve the response predictability by up to almost 100%.

After performing logistic regression, we were able to affirm that there is a significant relationship between the three variables. However, probably due to the sample size, the ratio strength of each had poor precision (wide confidence intervals).

Other authors relate pCR to the size of the tumor [32], the lymph node involvement [32], or the degree of differentiation [2,16,31,33]. High-grade tumors, as they have a high mitotic index, generally respond better to PST, but their value as an independent predictor of response to neoadjuvant treatment is not as clear [34]. In our study, we did not find significant differences.

In addition to these predictors usually available in daily clinical practice, there are many others that are still in the research phase: HER2DX, a supervised learning algorithm incorporating tumor size, nodal staging, and four gene expression signatures tracking immune infiltration, tumor cell proliferation, luminal differentiation, the expression of the HER2 amplicon, into a single score [32], the value of tumor-infiltrating lymphocytes (TILs) [35], the oncogene bcl-2 [36], HER2/CEN17 ratio [37], androgen receptor [38], and even Machine Learning with MRI Radiomics [39].

## 5. Conclusions

Most cases in our population of HER2 BC patients achieved PCR after PST, but there remains a significant percentage with residual tumor. For surgery planning, it is of great interest to have accessible clinical tools that allow us to distinguish between both, in order to be able to offer more conservative surgeries in pCR cases, especially in this type of tumors, which usually have accompanying microcalcifications that do not disappear even if the tumor does.

Our study shows that it is possible to predict complete or incomplete pathological response to PST adequately from the value of Ki67, the HER2 subtype, and the degree of radiological response. In HER2/HR− tumors, it is sufficient to know the degree of radiological response (MRI) for an adequate prediction. However, in the case of HER2 luminal tumors, the consideration of the Ki67 value significantly improves the predictive ability of MRI, as the degree of radiological response in these cases has lower diagnostic accuracy.

The limitations of our study include the retrospective design and a small sample size, which did not allow us to include more variables in the regression models. Therefore, future prospective studies are needed, with a bigger population and external validation of the predictive model, to confirm its clinical utility.

## Figures and Tables

**Figure 1 cancers-15-02757-f001:**
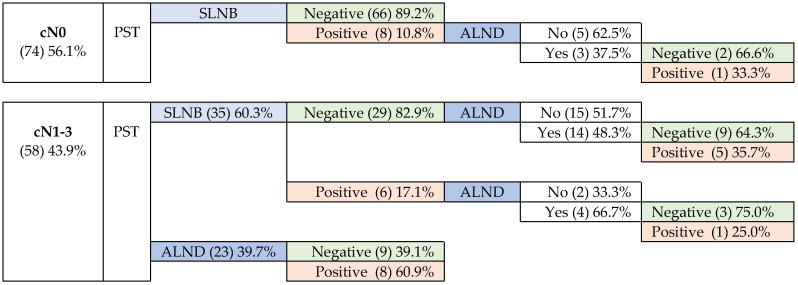
Management of patients according to initial axillary involvement.

**Figure 2 cancers-15-02757-f002:**
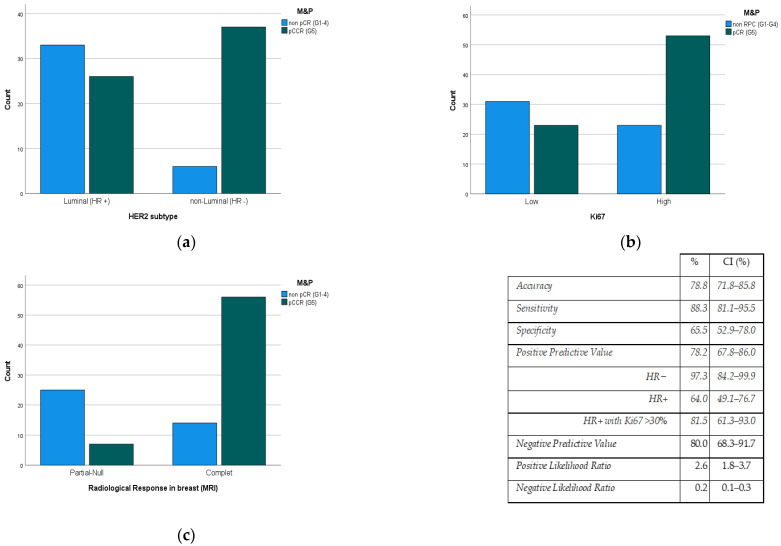
Pathological response degree (M&P criteria) to PST according to (**a**) HER2 subtype, (**b**) Ki67 value, and (**c**) radiological response rate (MRI). M&P: Miller and Payne criteria, PST: Primary systemic therapy, pCR: Pathological Complete Response; CI: Confidence Interval, HR: Hormonal Receptor, MRI: Magnetic Resonance Imaging.

**Table 1 cancers-15-02757-t001:** Patient and disease characteristics I.

	Total					M&P 1–4				M&P 5 (pCR)				
	n	Mean	SD	Min	Max	Mean	SD	Min	Max	Mean	SD	Min	Max	*p*
Age at diagnosis (y)	132	52.9	13.3	24	81	53.2	13.2	25	79	52.6	13.5	24	81	0.812
	**n**	**Median**	**IQR**	**Min**	**Max**	**Median**	**IQR**	**Min**	**Max**	**Median**	**IQR**	**Min**	**Max**	* **p** *
MRI Main tumoral size (mm)	132	29.5	20.0–40.0	9	100	28.0	20–40	11	110	30.0	20–40	11	100	0.805
MRI Total tumoral size (mm)	132	35.0	25.0–60.0	9	110	32.0	22–60	9	100	38.0	26–56	11	110	0.615
Ki67 (%)	130	37.5	25.0–51.3	4	95	30.0	24.8–40	5	90	40.0	30–60	4	95	0.006
Pathological tumoral size (mm)	132	0.0	0.0–8.8	0	70	11.0	5–17	0.1	70	0.0	0.0–0.0	0	15	0.000
Radiologic Response Rate (%)	132	100.0	95.0–100.0	−10	100	95.0	70–100	−10	100	100.0	100–100	0	100	<0.001
RCB (Pathologic response)	102	0.0	0.0–1.5	0	4.9	1.673	1.3–3.2	0.7	4.9	0.000	0.0–0.0	0	0	<0.001

y: years MRI: Magnetic Resonance Imaging; RCB: Residual Cancer Burden; IQR: Inter Quartile Range; M&P: Miller and Payne criteria; pCR: Pathological Complete Response; SD: Standard Deviation.

**Table 2 cancers-15-02757-t002:** Patient and disease characteristics II.

	All Patients		M&P 1–4		M&P 5 (pCR)		
	n	%	n	%	n	%	*p*-Value
cT	132	100.0	55	100.0	77	100.0	
1b	1	0.8	1	1.8	0	0.0	0.510
1c	35	26.5	15	27.3	20	26.0	
2	79	59.8	30	54.5	49	63.6	
3	17	12.9	9	16.4	8	10.4	
Focality	132	100.0	55	100.0	77	100.0	
Unifocal	80	60.6	36	65.5	44	57.1	0.335
Multifocal-multicentric	52	39.4	19	34.5	33	42.9	
Clinical node status (cN)	132	100.0	55	100.0	77	100.0	
N0	74	56.8	30	54.5	45	58.4	0.564
N1	30	22.0	14	25.5	15	19.5	
N2	26	19.7	11	20.0	15	19.5	
N3	2	1.5	0	0.0	2		
Histology of the tumor	132	100.0	55	100.0	77	100.0	
Ductal	123	93.2	49	89.1	74	96.1	0.115
Other	9	6.8	6	10.9	3	3.9	
Histological grade	132	100.0	55	100.0	77	100.0	
G1–G2	58	38.6	29	52.7	29	37.7	0.086
G3	74	56.1	26	47.3	48	62.3	
Ki67 value	130	100.0	54	100.0	76	100.0	
Low (≤30%)	54	41.6	31	22.2	23	30.2	<0.001
High (>30%)	76	58.5	23	42.6	53	69.7	
HER2 positive subtype	132	100.0	55	100.0	77	100.0	
HR− (non-luminal)	53	40.2	9	16.4	44	57.1	<0.001
HR+ (luminal)	79	59.8	46	83.6	33	42.9	
Radiological response after PST	132	100.0	55	100.0	77	100.0	
Complete	87	65.9	19	34.5	68	88.3	<0.001
Partial-Null	45	34.1	36	64.5	9	11.7	
Sentinel Node Biopsy	132	100.0	55	100.0	77	100.0	
No	23	17.4	16	29.1	7	9.1	0.003
Yes	109	82.6	39	70.9	70	90.9	
positive sentinel node	132	100.0	55	100.0	77	100.0	
0	95	87.2	29	74.4	66	94.3	0.01
1	13	11.9	9	23.1	4	5.7	
2	1	0.9	1	2.6	0	0.0	
>2	0	0.0	0	0.0	0	0.0	
Axillary involvement degree	14	100.0	10	100.0	4	100.0	
Macrometastases ^1^	7	50.0	7	70.0	0	0.0	
Macrometastases	7	50.0	3	30.0	4	100.0	
TAD	132	100.0	55	100.0	77	100.0	
No	120	90.9	51	92.7	69	90.9	0.539
Yes	12	9.1	4	7.3	8	9.1	
ALND	132	100.0	55	100.0	77	100.0	
No	87	65.9	28	50.9	59	76.6	0.02
Yes	45	34.1	27	49.1	18	23.4	
positive lymph nodes	132	100.0	55	100.0	77	100.0	
0	24	53.3	9	33.3	15	83.3	0.008
1–3	14	31.1	11	40.7	3	16.7	
4–9	4	8.9	4	14.8	0	0.0	
>9	3	6.7	3	11.1	0	0.0	
Pathologic response in breast (M&P)	132	100.0	55	100.0	77	100.0	
G1	6	4.5			55	41.7	
G2	0	0.0					
G3	25	18.9					
G4	24	18.2					
G5	77	58.3	77	58.3			
Pathologic response (RCB)	132	100.0	55	100.0	77	100.0	
pCR	59	57.8	0	0.0	59	76.6	
I	13	12.7	11	20.0	2	2.6	
II	23	22.5	21	38.2	2	2.6	
III	7	6.9	7	12.7	0	0.0	
Pathologic response in axilla (Sataloff)	64	100.0	31	100.0	33	100.0	
N-A	22	37.9	2	6.5	20	60.6	<0.001
N-B	14	20.7	6	19.4	8	24.2	
N-C	22	31.0	18	58.1	4	12.1	
N-D	6	10.3	5	16.1	1	3.0	

^1^ One patient had both macro and micrometastases. PST: Primary systemic therapy, pCR: Pathological Complete Response, TAD: Targeted axillary dissection, M&P: Miller and Payne criteria: RCB: Residual Cancer Burden.

## Data Availability

Data supporting these reported results can be found in the present article.

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
