# Peer review of "Surgical Interest of an Accurate Real-World Prediction of Primary Systemic Therapy Response in HER2 Breast Cancers"

_cancers, 2023, doi:10.3390/cancers15102757_

Round 1

Reviewer 1 Report

Title: Surgical interest of an accurate real-world prediction of primary systemic therapy response in HER2 breast cancers.

Review Report:

The article focuses on the importance of an accurate prediction of primary systemic therapy (PST) response in HER2-positive breast cancers. This is particularly relevant because HER2-positive breast cancers typically respond well to neoadjuvant treatment, which can help guide surgical decisions. The authors conducted a retrospective observational study on a population of 132 patients with HER2-positive breast cancer.

Strengths:

The study includes a well-defined patient population and a clear objective: to describe the degree of pathological response of HER2 breast cancer after PST and investigate its relationship with clinicopathological factors and MRI findings.

The authors employed a comprehensive range of variables, including age, ethnicity, tumor size, histological differentiation, hormonal receptors expression, Ki67 value, initial axillary clinical involvement, radiological response, and various surgical factors.

The study uses well-established criteria such as Miller and Payne and Residual Cancer Burden for evaluating pathological response.

Weaknesses:

Please add more discussion and conclusion to emphasize the importance of this study and its implications for future research

Please add more background information about different types of breast surgery and axillary lymph node dissection (ALND) (please cite: 1. Efficacy of da Vinci robot-assisted lymph node surgery than conventional axillary lymph node dissection in breast cancer - A comparative study. Int J Med Robot. doi: 10.1002/rcs.2307.  2. Robot-Assisted Minimally Invasive Breast Surgery: Recent Evidence with Comparative Clinical Outcomes. J Clin Med. doi: 10.3390/jcm11071827.)

The article does not mention any external validation of the predictive model, which could be useful in assessing its performance in different populations or clinical settings.

Overall, the article presents an interesting study on the importance of accurate real-world prediction of PST response in HER2 breast cancers. The authors identified Ki67 value, hormonal receptors condition, and MRI findings as significant predictors of pathological complete response, which can impact surgery planning. However, the study's limitations, including its retrospective design and small sample size, should be considered when interpreting the results. Future research could benefit from larger sample sizes, prospective designs, and external validation of the predictive model to further support its clinical utility.

Author Response

Point1 Please add more discussion and conclusion to emphasize the importance of this study and its implications for future research

Response1 Has been add this in: 

1 Introduction

"The aim of this study is to find which of the factors available in routine clinical practice can accurately predict, in HER2 BC, pCR after PST so that we can translate these results to surgical planning in de-escalation mode, both in the breast and in the axilla, without missing oncologic safety." 

Sub-head 4.7

"Properly identifying, from the clinical variables available at the time of surgical planning, the subgroup of patients with high pCR rates would allow us to correctly indicate less extensive surgeries both in breast and axilla [5], even when breast microcalcifications persist. This is of great importance in order not to overtreat surgically after PST."

5 Conclusions

"Most cases in our population of HER2 BC patients achieved PCR after PST, but there remains a significant percentage with residual tumor. For surgery planning, it is of great interest to have accessible clinical tools that allow us to distinguish between both, in order to be able to offer more conservative surgeries in pCR cases, especially in this type of tumors, which usually have accompanying microcalcifications that do not disappear even if the tumor does."

"Our study shows that it is possible to predict complete or incomplete pathological response to PST adequately from the value of Ki67, the HER2 subtype, and the degree of radiological response. In HER2/HR- tumors, it is sufficient to know the degree of radiological response (MRI) for an adequate prediction. But, it is in the case of HER2 luminal tumors where the consideration of the Ki67 value significantly improves the predictive ability of MRI, as the degree of radiological response in these cases has lower diagnostic accuracy"

Point2 Please add more background information about different types of breast surgery and axillary lymph node dissection (ALND) (please cite: 1. Efficacy of da Vinci robot-assisted lymph node surgery than conventional axillary lymph node dissection in breast cancer - A comparative study. Int J Med Robot. doi: 10.1002/rcs.2307.  2. Robot-Assisted Minimally Invasive Breast Surgery: Recent Evidence with Comparative Clinical Outcomes. J Clin Med. doi: 10.3390/jcm11071827.)

Response2 We do not consider the reference to robot-assisted surgery essential in this article, but if the reviewer/editor does, we will add it.

Point3 The article does not mention any external validation of the predictive model, which could be useful in assessing its performance in different populations or clinical settings.

Response3 Has been add this in

5 Conclusions:

"The limitations of our study include retrospective design and a small sample size, which did not allow us to include more variables in the regression models. Therefore, future prospectives researches are needed, with a bigger population and external validation of the predictive model, to confirm its clinical utility."

Reviewer 2 Report

Important review. Too much information some of it is repeated, I think the paper can be a shorter version and better highlight the key results.

In the introduction, they propose a better way to assess the response and select appropriate surgery. I don't think this is very clear in the conclusion.

English has to be improved especially in the introduction

Author Response

Point 1 Important review. Too much information some of it is repeated, I think the paper can be a shorter version and better highlight the key results.

Response1 The document has been fully reviewed. Other reviewer  have asked not only not to summarize, but to expand on some points.

Point 2 In the introduction, they propose a better way to assess the response and select appropriate surgery. I don't think this is very clear in the conclusion.

Response2

In the Introduction have been redrafted the aim of this study to clarify this point.

"The aim of this study is to find which of the factors available in routine clinical practice can accurately predict, in HER2 BC, pCR after PST so that we can translate these results to surgical planning in de-escalation mode, both in the breast and in the axilla, without missing oncologic safety." 

The conclusions have been redrafted to clarify this point.

"Most cases in our population of HER2 BC patients achieved PCR after PST, but there remains a significant percentage with residual tumor. For surgery planning, it is of great interest to have accessible clinical tools that allow us to distinguish between both, in order to be able to offer more conservative surgeries in pCR cases, especially in this type of tumors, which usually have accompanying microcalcifications that do not disappear even if the tumor does. 

Our study shows that it is possible to predict complete or incomplete pathological response to PST adequately from the value of Ki67, the HER2 subtype, and the degree of radiological response. In HER2/HR- tumors, it is sufficient to know the degree of radiological response (MRI) for an adequate prediction. But, it is in the case of HER2 luminal tumors where the consideration of the Ki67 value significantly improves the predictive ability of MRI, as the degree of radiological response in these cases has lower diagnostic accuracy.

 The limitations of our study include retrospective design and a small sample size, which did not allow us to include more variables in the regression models. Therefore, future prospectives researches are needed, with a bigger population and external validation of the predictive model, to confirm its clinical utility."

Point 3 English has to be improved especially in the introduction

Response3 The document has been fully reviewed

Reviewer 3 Report

The article discusses the increasing use of primary systemic therapy (PST) in the treatment of breast cancer, specifically in HER2-positive tumors. The benefits of PST include providing information on treatment effectiveness, having prognostic value, and allowing for less aggressive surgical treatments. The article notes that tumors with higher rates of response to PST, in terms of reduced tumor burden, are typically more aggressive. HER2-positive breast cancers are of particular interest as target therapies have been developed in recent years that can offer very effective treatments with controlled toxicity.

The article also notes that the best way to predict the pathological response to PST is through imaging techniques, especially magnetic resonance imaging (MRI), although these results are far from optimal. Therefore, it is of great interest to identify other predictors of PST response that might be involved in HER2 BC.

Overall, the article provides a good overview of the use of PST in the treatment of breast cancer, specifically in HER2-positive tumors, and highlights the need for further research to identify additional predictors of PST response.

Article need some grammar review. and check the margins (tables looks swifted to one of the documents margins)

Need more information:

Inter-rater reliability: If multiple researchers were involved in data collection or scoring of the HER2 status or other variables, it would be important to report the inter-rater reliability or any steps taken to ensure consistency

Statistical methods: The statistical methods used were appropriate for analyzing the data and determining the relationships between the variables. While the statistical methods described seem appropriate, more details on the exact tests used and any assumptions made should be provided.

I did not identify any significant grammar issues in the previous sections. However, there are a few minor errors or awkward phrases that could be improved here.

Line 93: cN1 who were negativized" could be changed to cN1 patients who became negative.

Author Response

Point 1. Article need some grammar review, and check the margins (tables looks swifted to one of the documents margins)

Response 1 the document has been fully reviewed orthographically and grammatically

Point 2 Need more information:

Inter-rater reliability: If multiple researchers were involved in data collection or scoring of the HER2 status or other variables, it would be important to report the inter-rater reliability or any steps taken to ensure consistency

Response 2. Data collection, as well as the evaluation of HER2 status and other immuno-histopathological variables, were performed, with common criteria, by 2 clinicians and 1 pathologist, respectively.

Point 3. Statistical methods: The statistical methods used were appropriate for analyzing the data and determining the relationships between the variables. While the statistical methods described seem appropriate, more details on the exact tests used and any assumptions made should be provided.

Response 3 Another reviewer has asked us to be more specific and less extensive, but we will be happy to provide the reviewer with any clarification in this regard.  

Point 4. There are a few minor errors or awkward phrases that could be improved here. Line 93: cN1 who were negativized" could be changed to cN1 patients who became negative.

Response 4. The document has been fully reviewed orthographically and grammatically and Line 93 changed.

Round 2

Reviewer 1 Report

strongly suggest for publication.